# Evolution, Expression and Meiotic Behavior of Genes Involved in Chromosome Segregation of Monotremes

**DOI:** 10.3390/genes12091320

**Published:** 2021-08-26

**Authors:** Filip Pajpach, Linda Shearwin-Whyatt, Frank Grützner

**Affiliations:** School of Biological Sciences, The University of Adelaide, Adelaide, SA 5005, Australia; filip.pajpach@adelaide.edu.au (F.P.); linda.shearwin@adelaide.edu.au (L.S.-W.)

**Keywords:** Aurora kinase, chromosome passenger complex, cohesin, monotreme, meiosis, sex chromosome multiple

## Abstract

Chromosome segregation at mitosis and meiosis is a highly dynamic and tightly regulated process that involves a large number of components. Due to the fundamental nature of chromosome segregation, many genes involved in this process are evolutionarily highly conserved, but duplications and functional diversification has occurred in various lineages. In order to better understand the evolution of genes involved in chromosome segregation in mammals, we analyzed some of the key components in the basal mammalian lineage of egg-laying mammals. The chromosome passenger complex is a multiprotein complex central to chromosome segregation during both mitosis and meiosis. It consists of survivin, borealin, inner centromere protein, and Aurora kinase B or C. We confirm the absence of Aurora kinase C in marsupials and show its absence in both platypus and echidna, which supports the current model of the evolution of Aurora kinases. High expression of *AURKBC*, an ancestor of *AURKB* and *AURKC* present in monotremes, suggests that this gene is performing all necessary meiotic functions in monotremes. Other genes of the chromosome passenger complex complex are present and conserved in monotremes, suggesting that their function has been preserved in mammals. Cohesins are another family of genes that are of vital importance for chromosome cohesion and segregation at mitosis and meiosis. Previous work has demonstrated an accumulation and differential loading of structural maintenance of chromosomes 3 (SMC3) on the platypus sex chromosome complex at meiotic prophase I. We investigated if a similar accumulation occurs in the echidna during meiosis I. In contrast to platypus, SMC3 was only found on the synaptonemal complex in echidna. This indicates that the specific distribution of SMC3 on the sex chromosome complex may have evolved specifically in platypus.

## 1. Introduction

The process of chromosome segregation involves a large number of components and requires tight regulation. This is mediated by many different proteins, including centromere and kinetochore proteins, cohesins, Aurora kinases, and proteins of the chromosome passenger complex (CPC). We now have an increasing understanding of how all these proteins work together to ensure faithful chromosome segregation. However, very little is known about segregation of complex chromosome multiples at meiosis and how these proteins might be involved in their segregation. One of the most striking examples of a chromosome multiple is the sex chromosome system of the oldest surviving mammals—the monotremes. In platypus males, there are 5X and 5Y chromosomes (5X and 4Y chromosomes in echidna males) that form a complex chain multiple at meiosis [1]. This chain multiple has been described and investigated in detail previously, and interestingly, it exhibits some unique properties that distinguish it from sex chromosome multiples in other organisms [2]. Since very little is known about genes involved in chromosome segregation of monotremes, and given the phylogenetic position of monotremes and their extraordinary sex chromosome system, it is important to investigate the genes involved in segregation in this mammalian lineage. Indeed, it is possible that the evolution of this complex sex chromosome system may have led to changes in segregation apparatus.

The Aurora kinases (AURKs) are members of an evolutionarily conserved family of serine/threonine kinases which have many essential roles in regulating chromosome segregation at both mitosis and meiosis. There are three isoforms of AURKs—many organisms express AURKA and AURKB isoforms, while eutherian mammals also express the germ-specific isoform AURKC [3]. These isoforms share a substantial sequence similarity and have a very similar protein structure, consisting of regulatory motifs on both C and N-terminal ends, as well as the central catalytic domain between them. The remarkable difference is found in the case of AURKC, which lacks a considerable portion of the N-terminus found in the other two isoforms, including the KEN (K-E-N amino-acids containing) and D-box activating motifs. The *AURKA* gene is the oldest gene and is found alongside the ancestral *AURKBC* gene in the cold-blooded vertebrates [4]. In birds, the *AURKA* gene is still present, but the *AURKBC* gene was lost. However, this is not detrimental, since Aurora kinases are highly interchangeable—for example, AURKA from starfish can compensate for functions of both human AURKA and AURKB if those are knocked-down [5], and robust compensatory capabilities have also been proved for AURKA in mouse meiosis [6]. In monotremes and marsupials, there are *AURKA* and *AURKBC* genes [3], indicating that *AURKC* gene has only been acquired in eutherian mammals by duplication and subsequent diversification of the ancestral *AURKBC* gene.

The nuclear localization and functions of AURKA and AURKB are known in mitosis, where these kinases regulate spindle dynamics, chromosome condensation and cohesion, kinetochore-microtubule attachments, error correction of this attachment, spindle assembly checkpoint, and cytokinesis [7]. Thus, AURKs are clearly involved in the regulation of chromosome segregation. However, very little is known about expression or functions of AURKs when it comes to meiosis—some studies have confirmed that, for instance, AURKA together with Polo-like kinase 1 (PLK1) is required for the correct migration of duplicated centrosomes at male meiosis I and II in order to form a bipolar spindle [8,9]. Furthermore, it is unknown why there is the additional isoform AURKC present in germ cells of eutherian mammals, or why it is not present in marsupials. Moreover, the existence of AURKC in monotremes has not been proved so far. Therefore, it is important to assess the evolutionary status of AURKs in monotremes in order to generate a more complete picture of the AURK evolution in mammals.

AURKA is localized near centrosomes at mitosis and meiosis and acts on its own, while either AURKB or AURKC are a part of the CPC, along with borealin (CDCA8), survivin (BIRC5), and inner centromere protein (INCENP) [10]. Previous research suggests that INCENP is required to recruit survivin to form the CPC, and in turn, survivin recruits AURKB [11]. This is complemented by borealin, which has a role in targeting these proteins too, as well as being responsible for stability of the spindle [12]. CPC is an important regulatory element that mediates correct chromosome segregation—this is achieved by cooperation with a large pool of other proteins involved in chromosome segregation (Figure 1). This includes centromeric and kinetochore components like histone variant CENP-A, constitutive centromere-associated network (CCAN) of kinetochore and KMN complex that is composed of complexes KNL1, MIS12, and NDC80 [13,14].

In addition to the CPC, structural maintenance of chromosomes (SMC), which contains the cohesins, is a group of critical proteins that not only mediate sister chromatid cohesion during mitosis and meiosis but are also involved in the synapsis of homologous chromosomes at meiosis. Cohesins are protein complexes that are composed of two SMC proteins (always SMC3 and either mitosis-specific SMC1A or meiosis-specific SMC1B), a kleisin (e.g., RAD21 or REC8), and an SCC component (mitosis-specific SA-1/STAG1 or SA-2/STAG2 or meiosis-specific STAG3) [15,16]. These proteins are critical for mitosis, where they ensure cohesion of sister chromatids until the moment they segregate at anaphase. At meiosis, cohesins are not only responsible for sister chromatid cohesion, but are also required for homolog interactions and initiation of the assembly of synaptonemal complex [16,17,18]. It was previously described that SMC3 behaves in an unusual way during platypus meiosis and that this might be related to organization and segregation of the sex chromosome multiple [19]. Apart from the study of Casey et al., there is practically no research investigating the segregation-related proteins in monotremes or more generally the segregation of chromosome multiples at meiosis.

Our analysis of these key components in the basal lineage of monotremes confirmed that *AURKA* and *AURKBC* are present, but *AURKC* is not, providing further clarification for the evolution of this gene family. Additionally, the expression and immunofluorescence analysis provide more information about the segregation-related proteins in monotremes. Interestingly, comparison of meiotic distribution of the cohesin SMC3 shows that the accumulation observed in platypus is absent at echidna meiosis. This may be an indication that organization and segregation of the sex chromosome multiple is different between otherwise closely related species, raising potential implications for the cohesin involvement in the segregation of complex chromosome multiples.

## 2. Materials and Methods

### 2.1. Bioinformatic Analysis, Transcriptome Mapping, and Expression Analysis

All genome database search was carried out on the NCBI website. Available amino-acid sequences were obtained from NCBI. The NCBI conserved domains and pBLAST tools were used for sequence similarity searches. The terminal tnBLAST tool was used to search for the new sequences. Amino acid sequence alignments were carried out using Geneious software (v11.1.4) with the following settings: ClustalW alignment, BLOSUM, gap open cost = 10, gap extend cost = 0.1. Phylogenetic trees were constructed using Geneious software (v11.1.4) with the following settings: RAxML, γ BLOSUM 62, rapid hill-climbing, number of starting trees or bootstrap replicates = 1, parsimony random seed = 1, and zebrafish was used as an outgroup. GenScan online tool (http://hollywood.mit.edu/GENSCAN.html (accessed on 13 November 2020)) was used to predict ORFs with the default settings.

Transcriptome mapping was carried out in terminal using tools hisat2 (v2.1.0) and samtools (v1.9) on transcriptome datasets (.fastq files) that have been previously obtained by RNA sequencing and processed bioinformatically (includes concatenation, adapter removal, and quality control)—availability of datasets is described in the data availability section. Mapped transcripts were visually examined, and their sequences extracted from IGV software (v2.7.2).

Expression analysis was performed in terminal using subread featureCounts tool (v2.0) by quantifying platypus single-end reads (generated as described in the previous paragraph) mapping to the newest version of platypus genome publicly available on NCBI (mOrnAna1.pri.v4). Each tissue transcriptome represents two samples (one male and one female) excepting ovaries (two females) and testes (three males). Expression charts were produced in RStudio (v1.3.1093), and error bars represent standard deviation of mean RPKM.

### 2.2. Cell Lines

Established platypus fibroblast cell line collected under the AEC permit S-032-2008 was used [20]. Echidna fibroblast cell line was established de novo from a roadkill male echidna using previous methodology [21].

### 2.3. Preparation of Meiotic Cells

Platypus and echidna testicular material in 10% DMSO (AEC permit S-032-2008) was applied onto a slide and flooded with CSK buffer (containing 100 mM NaCl, 100 mM sucrose, 0.5% Triton X-100, 3 mM MgCl_2_, and 10 mM PIPES, pH = 6.8) for 10 min, then flooded with 4% paraformaldehyde/1× PBS for 10 min and rinsed in 1× PBS. Slides were used immediately.

### 2.4. Immunostaining

Immunostaining was performed following standard procedures [19] using 0.5% Triton X-100/1× PBS for permeabilization for 20 min, 5% *v*/*v* goat serum/1% BSA/1× PBS as a blocking buffer for 1 h in total; anti-AURKB (ab2254, abcam, Melbourne, Australia), anti-INCENP (ab12183, abcam), and anti-SMC3 (ab244287, abcam) primary antibody in the blocking buffer (1:200 dilution); and a fluorescent secondary antibody (ab150080 or ab150077, abcam) in the blocking buffer (1:400 dilution). Chromosomes were counterstained using DAPI. Images were taken using a Nikon Eclipse Ti microscope equipped with achromatic objectives and a Nikon DS-Qi1Mc camera and NIS-Elements AR software (version 4.20).

## 3. Results

### 3.1. Analysis of Monotreme AURK Genes

To date *AURKA* and *AURKB* genes have been found in monotremes—the information about *AURKC* is missing. Utilizing newly assembled genomes of both platypus and echidna [20], we revisited questions about the presence or absence of *AURKC*. The alignment of AURK amino acid sequences from different animal groups was performed, showing high sequence similarity. This approach yielded a highly conserved amino acid sequence. We used this sequence, as well as the consensus sequence of the conserved STKc-AURKB-like domain of the human AURKC, to perform translated nucleotide BLAST (tnBLAST) against the platypus and echidna genomes. This resulted in sequence matches on chromosome 21. Reciprocal BLAST with these matches against the current human genome, as well as open reading frame (ORF) prediction and a subsequent protein BLAST (pBLAST), returned the same match on chromosome 21, corresponding to testis-specific Serine/Threonine kinase 1 (TSSK1). Next, synteny analysis was performed to find a syntenic region in platypus to the human chromosome 19 region containing *AURKC*. We identified genes epsin 1 (*EPN1*) and tripartite motif containing 28 (*TRIM28*) surrounding *AURKC* in human that mapped to a common region in platypus and echidna. Hence, we used this region to predict all amino acid sequences using GenScan. The subsequent pBLAST of these sequences did not produce any similarity to AURKC. Taken together, our results suggest that *AURKC* is not present in monotreme genomes. Synteny analysis was also performed for *AURKA* and *AURKB*, showing a perfect (*AURKA*) or nearly perfect (*AURKB*) conservation of synteny between human, opossum, and monotremes (Figure 2).

Alignment of the AURK proteins from the three mammalian lineages, represented by human (eutherian), opossum (metatherian), and platypus and echidna (prototherian) (Figure 3), shows the high degree of similarity between all AURK proteins, with the highest similarity over 70% seen within the kinase domain. The alignment of all AURKs of different vertebrates shows clustering of respective AURKs (Figure 4a). The monotreme AURKs are highly similar to each other, with the AURKAs and AURKBCs between platypus and echidna sharing over 90% sequence identity. The monotreme AURKAs and AURKBCs are at least 75% identical to their opossum equivalents. The alignment shows that the phosphorylation site required for the conformational change to enable kinase domain activity is conserved in all AURKs. Cell cycle regulation of AURK proteins is very important for cell cycle progression and Short, Linear Motifs (SLiMs) in AURKs are degrons required for proteasome-mediated degradation. Degradation is mediated by the anaphase-promoting complex/cyclosome (APC/C), a ubiquitin-ligase activated by the WD repeat containing Fizzy family proteins Cdc20 and Cdh1. Cdc20-APC/C and Cdh1-APC/C target destruction of proteins carrying the D-Box, and Cdh1-APC targets those carrying the KEN box [22,23,24,25,26,27].

The D-Box within the C-terminal region of the kinase domain appears to be conserved in the monotreme AURKs. Notably, the KEN box, although present in the monotreme AURKBC proteins, is missing in both the platypus and echidna AURKA proteins. The D-Box-activating (DAD/A) sequence is conserved in the monotreme AURKAs, as in the human and opossum AURKAs, and is an atypical degron required for APC/C-mediated degradation [28,29]. The presence of the conserved degrons in the monotreme AURKs indicate their levels are likely controlled throughout the cell cycle via the APC/C degradation system.

Recent publication of the first echidna genome [20] enabled comparative analysis of monotreme *AURKBC*—this gene was mapped to echidna chromosome Y4. However, because AURKBC is likely critical for chromosome segregation in females too, we mapped the echidna ovary transcriptome to the echidna genome and found that AURKBC transcripts were also present in females. Moreover, platypus *AURKBC* is located upstream of genes that have been previously assigned to the pseudo-autosomal region (PAR) of chromosome X5 (for example flotillin 1 (*FLOT1*)), suggesting that platypus *AURKBC* also localizes to this PAR. Because platypus PAR of X5 is syntenic to echidna PAR of X4 [20], this suggests that echidna *AURKBC* would be located in PAR of Y4/X4.

Because monotremes lack *AURKC*, we expected that *AURKBC* would be highly expressed in testes and ovaries instead, as opposed to more universally expressed *AURKA*. Therefore, we quantified the expression of platypus *AURKA* and *AURKBC*, showing that both AURKA and AURKBC transcripts are preferentially produced in ovaries, testes, and fibroblasts compared to other examined tissues. The transcript levels of AURKA were generally higher, with an unexpectedly high ovary expression (Figure 4b). Lastly, in order to investigate the localization of AURKBC, we performed immunofluorescence on platypus- and echidna-cultured fibroblasts with an antibody against this protein. This revealed that during mitosis, AURKBC behaves in a way typical for other mammals (Figure 4c)—it has expected dispersed localization in an interphase nucleus, localizes predominantly to chromatin at metaphase, and at anaphase to telophase transition, it remains on the midbody between two newly forming daughter cells.

### 3.2. Analysis of CPC and Chromosome-Segregation-Related Proteins

During cell division, AURKB functions as a part of CPC, but any information on expression or meiotic behavior of CPC components in monotremes is missing. Therefore, we decided to investigate expression profiles and meiotic immunofluorescence patterns of other CPC members (BIRC5, CDCA8, and INCENP), as well as segregation-related proteins—nucleosome and inner kinetochore proteins CENP-A, B, and C; central kinetochore proteins CENP-S, T, and W; proteins of CCAN (i.e., CENP-H, I, K, L, M, N, O, P, Q, U); outer kinetochore proteins KNL1, MIS12, and NDC80; and cohesins REC8, SMC1A, SMC1B, SMC3, STAG1, STAG2, and STAG3 in platypus. In other mammals, some of these proteins act prominently at meiosis—they are expressed at low levels in various somatic tissues with an increased expression in testes, for example human SMC1B or STAG3. Other proteins act at both mitosis and meiosis; for example, in humans the expression of SMC3 is notably higher in testes and rapidly dividing tissues, such as bone marrow and lymph node, but otherwise the expression is uniform across most other tissues. Finally, some proteins are specific to mitosis, such as SMC1A in humans. We investigated expression patterns of these genes in transcriptomes of the platypus. Similar relative expression patterns of transcripts in platypus were observed which may correlate with protein levels: CPC transcripts BIRC5 and CDCA8 exhibited similar expression patterns to each other, while INCENP was more uniformly expressed across most tissues, except fibroblasts. Outer kinetochore transcripts MIS12 and NDC80 exhibited similar expression patterns to each other, while KNL1 appeared to be only expressed above baseline level in fibroblasts, ovary, and testis. All cohesins except for REC8 exhibited the expected expression patterns—meiosis-specific SMC1B and STAG3 had increased ovary and testis expression, while mitotic/meiotic SMC3 and mitosis-specific SMC1A, STAG1, and STAG2 were expressed more uniformly across tissues with increased expression in fibroblasts, ovary, and testis (Figure 5). The expression of REC8, which acts as a meiosis-specific cohesin as well as a recombination protein, was surprisingly low, but otherwise as expected—across all tissues with increased expression in ovary and testis (Figure A1). All of the examined genes exhibited a relatively increased or very high expression in fibroblasts, and some of them (such as CDCA8) had surprisingly high expression in ovary.

The relative expression patterns of almost all analyzed centromeric transcripts (CENPs) were mostly consistent with low expression across tissues but relatively high expression in fibroblasts, ovary, and testis, except for CENP-P with very high expression in ovary and CENP-T with testis-specific expression (Figure A1).

Out of all analyzed proteins, meiotic immunofluorescence was successfully performed for INCENP and SMC3. A comprehensive immunofluorescence analysis of SMC3 has been done for prophase I in platypus before [19], so we compared this to echidna. At prophase I in platypus, SMC3 is heavily loaded on the unpaired regions of sex chromosomes [19], but we did not observe this at echidna prophase I (Figure 6a,b). A reliable INCENP immunofluorescence was detected at prophase I, where a scattered nuclear pattern was observed for both platypus and echidna (Figure 6c,d).

## 4. Discussion

Numerous proteins are involved in the organization and segregation of chromosomes at mitosis and meiosis. While most of the key components and complexes involved in chromosome maintenance and segregation are evolutionarily highly conserved, some have undergone duplication, gene loss, and sub-functionalization. To gain a finer understanding of changes in those genes during mammalian evolution, we focused on the analysis of CPC and SMC genes in monotremes. Our data on monotreme AURKs support the previous understanding of the evolution of these proteins—AURKA is the most ancestral of the three kinases present in most organisms. In cold-blooded vertebrates, there is also an ancestral isoform, AURKBC, in addition to AURKA [10]. The AURKBC isoform was probably lost in birds, since it is found in most other reptiles. Because AURKs are highly interchangeable—for example, starfish AURKA can functionally compensate for either AURKA and AURKB in human [5]—this loss is most likely not detrimental. The same ancestral gene encoding AURKBC in cold-blooded vertebrates is also present in marsupials [3]. Only after eutherian mammals diverged did *AURKBC* undergo duplication and subsequent diversification, resulting in separate *AURKB* and *AURKC* genes in eutherian mammals [3]. Our data provide the missing information on monotremes, where *AURKA* and *AURKB* are both present, but *AURKC* is absent. This indicates that monotremes may still use the ancestral AURKBC isoform. Indeed, the expression profile of platypus AURKBC suggests that this isoform is highly expressed in testes and ovaries (similarly to the expression of gonad-specific AURKC in eutherian mammals). This suggests that monotreme AURKBC functions at meiosis much like in cold-blooded vertebrates and marsupials.

The genomic region containing *AURKA* and *AURKB* share synteny between monotremes, opossum, and human. A notable difference is found when it comes to the genomic location of *AURKBC* in monotremes—as opposed to other vertebrates, monotreme *AURKBC* is located on sex chromosomes rather than autosomes. However, the corresponding *AURKB*-bearing chromosomes (monotreme Y4/X5, opossum 2, and human 17) do show synteny between limited genomic regions. To our knowledge, it is also the only Aurora-kinase-coding gene localized on a sex chromosome in animals. Our data further suggest that monotreme *AURKBC* is located specifically in the PAR (shared region) and expressed in both sexes. The sex chromosome localization is still interesting from a dosage compensation and meiotic sex chromosome inactivation (MSCI) point of view. However, it appears that monotremes exhibit a global transcriptional down-regulation at meiotic pachytene rather than MSCI and lack dosage compensation [30,31].

The amino acid sequence of AURKs contains defined regulatory elements required for various functions of AURKs. Indeed, almost all of these elements are perfectly conserved between the three mammalian lineages. However, the KEN box, although present in the monotreme AURKBC proteins, is missing in both the platypus and echidna AURKA proteins. This indicates that, unlike other mammalian AURKAs, the monotreme AURKA proteins are degraded during the cell cycle in the absence of the KEN-Cdh1-APC/C directed pathway.

Our immunofluorescence result indicates that during mitosis, platypus AURKBC is localized similarly to AURKB of other mammals—it exhibits the scattered nuclear pattern at interphase, chromatin localization at metaphase, and midbody localization at anaphase. However, we could not obtain representative results for AURKBC during meiosis.

The expression profiles of other chromosome-segregation-related genes that we examined shared some similarities to other mammals, but also exhibited some surprising differences: expression profiles of CPC transcripts (AURKBC, BIRC5, CDCA8, and INCENP) indicate that these proteins are conserved in mammals, as they are expressed across both mitotically and meiotically dividing tissues, highlighting their essential role in chromosome segregation. Expression patterns of KNL1, MIS12, and NDC80 (all members of the outer kinetochore layer) were similar to what is seen in databases for human or mouse transcriptomes, suggesting their conservation among mammals. Cohesins (REC8, SMC1A, SMC1B, SMC3, STAG1, STAG2, STAG3) exhibited a typical expression where SMC1B and STAG3 are limited to meiosis and therefore are highly expressed in ovaries and testes but not in other tissues. Cohesin SMC3 is expressed in both mitotic and meiotic tissues, while REC8 is a meiotic cohesin but has additional functions in recombination, so their expression patterns are as expected. Centromeric proteins (CENPs) exhibited consistent expression patterns with uniform expression across tissues and increased expression in fibroblasts, ovary, and testes. Notable exceptions involve CENP-T, which appeared to be mostly limited to testicular expression, and CENP-P, with markedly increased ovary expression. To the best of our knowledge, there are no reported explanations of why these specific CENPs would exhibit such unusual expression (since they are parts of kinetochore complex at both mitosis and meiosis). Cultured fibroblasts present a rapidly dividing cell population with a rapid turnover of chromosome-segregation-related proteins, and therefore, most genes exhibited a dramatically high expression in this tissue. In ovaries, the expression was much higher than expected in the case of AURKA, BIRC5, CDCA8, CENP-A, CENP-P, NDC80, SMC1B, and SMC3, which may be due to the high division rate of the support cells in ovaries required for a successful oogenesis. Finally, some expression differences between proteins belonging to a common complex (i.e., INCENP compared to BIRC5 and CDCA8, and KNL1 compared to MIS12 and NDC80) cannot be easily explained by transcriptome analysis only. Expression levels of a given transcript may not reflect the protein level. Many genes presented in this study exhibit a relatively wide range of expression between the testes of three individual males (e.g., for KNL1, STAG2, STAG3, and most CENPs). Testes undergo major change in this seasonal breeder, and as samples were collected during breeding season, this may contribute to the observed differences. It would be of a great interest to generate transcriptome data from animals in and out of breeding season.

During prophase I, both platypus and echidna INCENP exhibits the scattered nuclear localization that seems to differ from that of mouse, where INCENP co-localizes with synaptonemal complex until late pachytene, after which it repositions into heterochromatic chromocenters [11]. In echidna, the INCENP signal is specific to the nucleus, but no clear strings that are typical for synaptonemal complex are visible. In platypus, it appears that the string pattern is present to a certain degree upon careful examination. However, it remains unclear what prophase I stage we observed, as we could not co-localize INCENP with SMC3 or synaptonemal complex proteins due to lack of working antibodies. Nevertheless, INCENP is one of the less conserved proteins from the CPC group (52.4% pairwise identity, Figure A2), which may explain the difference between the observed result and what has been reported before in mouse.

It has been previously reported that during platypus pachytene, there is a sex-chromosome-specific loading of SMC3, followed by the retraction of SMC3 into proteinaceous structures at diplotene [19]. Our results confirm this for platypus, but we did not observe a similar pattern in echidna during pachytene, indicating differences within the monotreme group. It was previously speculated that the differential SMC3 distribution could indicate a role in the organization and segregation of the sex chromosome complex [19]. It is therefore surprising that a similar distribution is not observed in echidna. However, the current study did not investigate other stages of prophase I, and it will be interesting to investigate the differences in SMC distribution in more detail. Indeed, the expression profile of platypus SMC3 does not show elevated testicular expression, suggesting that the observed SMC3 accumulation at prophase I does not depend on transcription. In platypus and echidna, eight of the ten sex chromosomes are homologous, indicating that a complex sex chromosome system existed in the monotreme ancestor. The differential accumulation on sex chromosomes may be platypus specific or has been lost in echidna after their divergence 55 million years ago.

## Figures and Tables

**Figure 1 genes-12-01320-f001:**
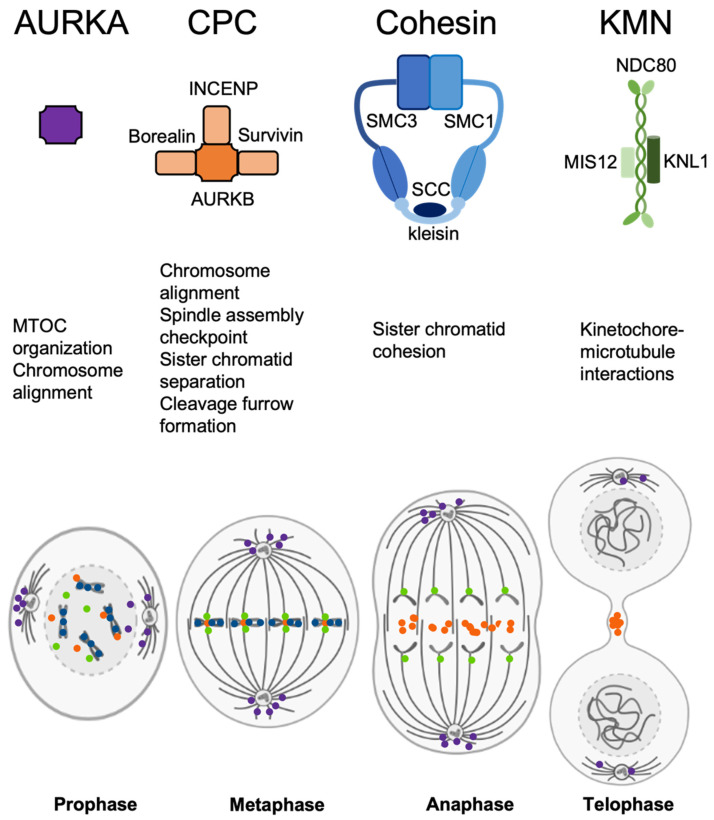
Summary of structure, main functions, and mitotic localization of the protein complexes analyzed in this study. Purple—AURKA, orange—CPC, blue—cohesin, green—KMN.

**Figure 2 genes-12-01320-f002:**
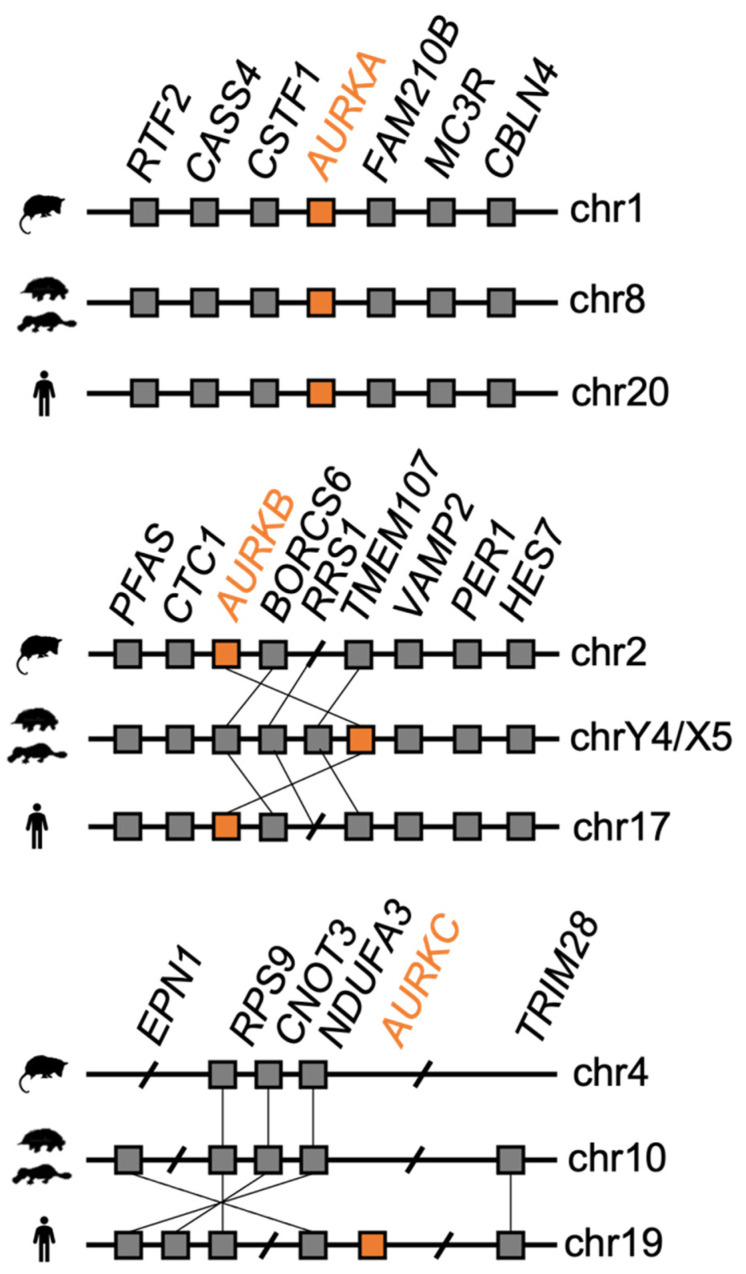
Synteny maps of genomic regions containing *AURKA*, *AURKB,* and *AURKC*, showing the conservation of synteny between respective chromosomes of human, opossum, and monotremes (platypus and echidna). Note the sex chromosome localization of the monotreme *AURKB* (*AURKBC*) and the absence of *AURKC* in monotremes. Orange square – gene of interest (*AURK*); Grey square – surrounding gene.

**Figure 3 genes-12-01320-f003:**
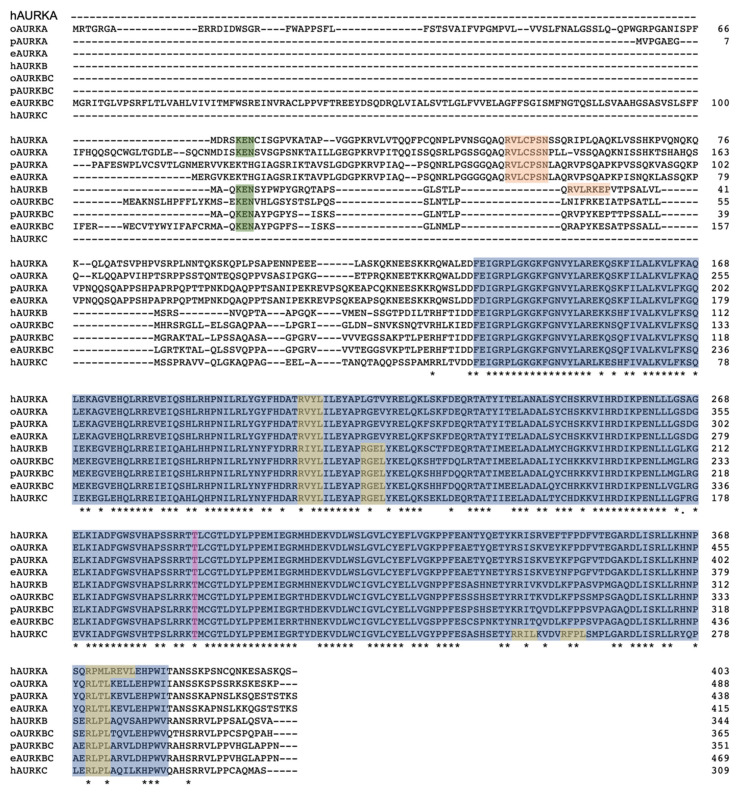
Alignment of AURK proteins of human (h), opossum (o), platypus (p), and echidna (e). The kinase domain is highlighted in blue; the degrons were identified by comparison with those of the human AURK proteins [29]; the KEN box in green; the DAD/A box in orange; the D-Box in yellow; and the Threonine, which becomes phosphorylated for enzyme activity, is highlighted in pink. *, fully conserved residue.

**Figure 4 genes-12-01320-f004:**
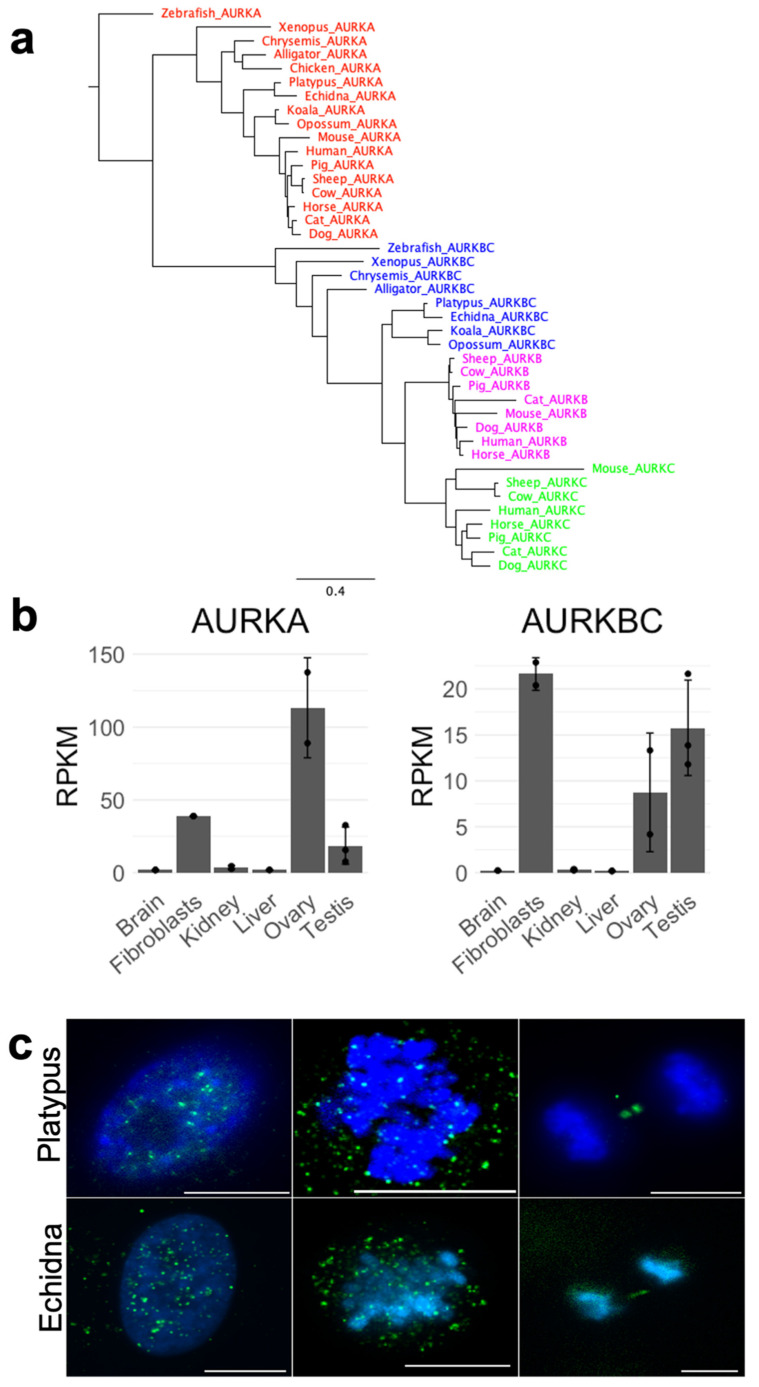
(**a**) Phylogenetic tree showing clustering of individual AURKs—AURKAs, AURKBCs (the ancestral version before AURKC emerged), AURKBs (therian mammals), and AURKCs (placental mammals only) cluster together, respectively. (**b**) The expression profile of platypus AURKA and AURKBC. (**c**) Immunofluorescence of platypus and echidna AURKBC in cultured fibroblasts, showing a similar localization to other mammals at mitosis (scattered interphase pattern, metaphase with predominant chromatin localization, and midbody-specific signal at anaphase, from left to right, respectively). Green signal—AURKBC, blue signal—DAPI. Scale bar—10 µm.

**Figure 5 genes-12-01320-f005:**
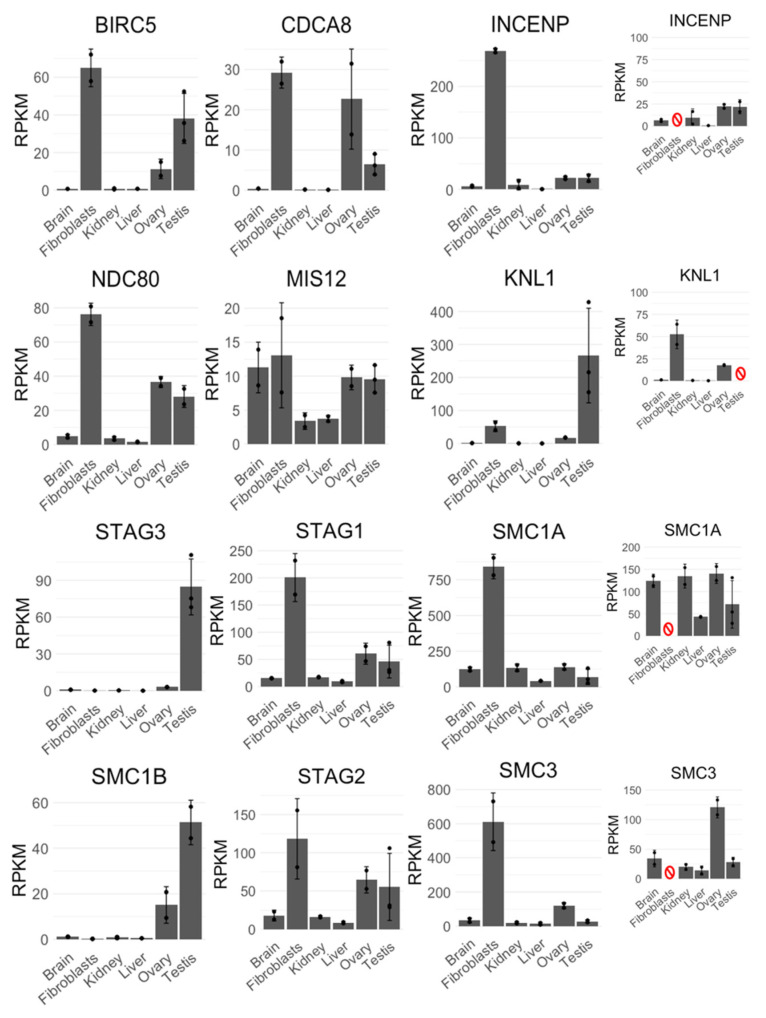
Expression profiles of CPC (BIRC5, CDCA8, INCENP), KMN complex (KNL1, MIS12, NDC80), and cohesins (STAG1—3, SMC1A, SMC1B, SMC3) obtained from single-end transcriptome datasets originating from RNA sequencing of different platypus tissues. Error bars represent standard deviation of mean RPKM calculated from two datasets per each tissue, except testes, where three sets were obtained. The rightmost expression profiles (INCENP, KNL1, SMC1A, and SMC3) also have smaller insets representing the data after removing (red symbol) the tissue with the highest RPKM value, which otherwise masks smaller RPKM values of other tissues in the original charts (i.e., testis for KNL1 and fibroblasts for INCENP, SMC1A, and SMC3).

**Figure 6 genes-12-01320-f006:**
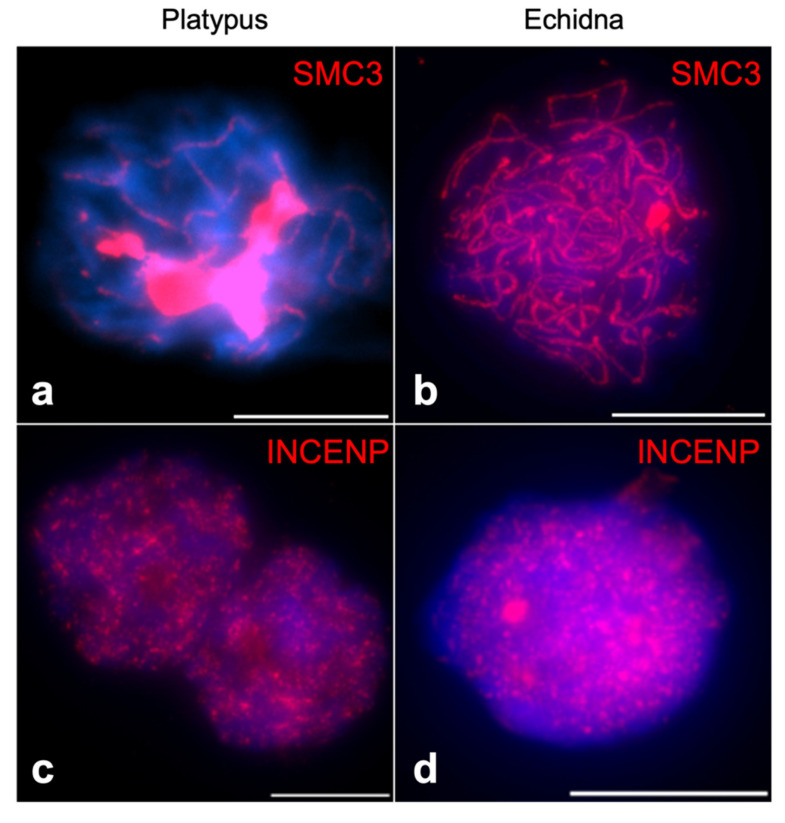
Immunofluorescence of (**a**) SMC3 of platypus spermatocyte at pachytene showing heavy loading of this protein on sex chromosomes, leaving autosomal axis barely visible, (**b**) SMC3 of echidna spermatocyte at pachytene showing no sex-chromosome-specific SMC3 accumulation. Immunofluorescence of (**c**) INCENP of platypus spermatocyte at prophase I and (**d**) INCENP of echidna spermatocyte at prophase I, both exhibiting scattered nuclear pattern. Red signal—SMC3 or INCENP as indicated, blue signal—DAPI. Scale bar—10 µm.

## Data Availability

Publicly available datasets were analyzed in this study. This data can be found here: https://www.ncbi.nlm.nih.gov (accessed on 15 September 2020) for all amino-acid sequences found in this study; https://www.ncbi.nlm.nih.gov/geo/query/acc.cgi?acc=GSE30352 (accessed on 24 February 2021) for the platypus transcriptome datasets.

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
