# Peer review of "Evolution, Expression and Meiotic Behavior of Genes Involved in Chromosome Segregation of Monotremes"

_genes, 2021, doi:10.3390/genes12091320_

Round 1
Reviewer 1 Report
This is a beautiful study about presence and distribution of some CPC and some cohesin proteins in platypus and echidna, and what could have been the evolution of some of these genes. They have demonstrated the absence of one of the AURK protein (AURKC) in both species studied as it occurs in marsupials. This fact supports the current model of the evolution of Aurora kinases. They also have tried to see if the expression of another AURK compensate the lack of AURKC showing that probably AURKBC could compensate it. They have also analyzed the expression of other proteins implied in cell division in platypus y different tissues, increasing the knowledge in this still unexplored area. Since the authors knew quite well the distribution of SMC3 in platypus, they have also analyzed the distribution of this protein in echidna, demonstrating some differences between both species, but they don´t explain the possible causes and more studies must be developed in this field.
It is an interesting study in not very well-known species. The strengths of this work are the beautiful analysis of genes and protein expression, and how they probe the absence of AURKC and how AURKBC compensate this lack. Also, I like how their results supports the evolution of AURK proteins and the manner they show it.
Nevertheless, there are some minor points that may improve their work:
As a general comment, once they demonstrated the absence of AURKC in both species, it would be desirable that they refer to it as AURKBC or explain why some times they refer to it as AURKB. I understand it for the immunolabelling, because they used an anti-AURKB antibody, but sometimes it is not very clear for me the difference. For example, in page 7 firs paragraph “Recent publication of the first echidna genome [18] enabled comparative analysis of monotreme AURKBC – this gene was mapped to echidna chromosome Y4. However, because AURKBC is likely critical for chromosome segregation in females too, we mapped 208 the echidna ovary transcriptome to the echidna genome and found that AURKB transcripts were also present in females. Moreover, platypus AURKB is located upstream of genes that have been previously assigned to the pseudo-autosomal region (PAR) of chromosome X5 (for example FLOT1), suggesting that platypus AURKB also localizes to this PAR. Because platypus PAR of X5 is syntenic to echidna PAR of X4 [18], this suggests that echidna AURKB would be located in PAR of Y4/X4. “
I understand how difficult it is to immunolabelling proteins in testes in those species and the great value of their results but, in my opinion the SMC3 results in echidna doesn´t have much to do with the other results. In fact, they haven´t discuss very much about it. I understand they use SMC3 to identify the different meiotic stages and that what they have found is an interesting result, but maybe it could be remove from the summary.
Related also with the immunodetection experiments, in relation with immunolabelling of AURKB in platypus and echidna, (Fig 4 C): In metaphase it is difficult to see which is the distribution of AURKB protein. The labelling appears as dots both on chromatin and on cytoplasm (maybe due to the difficulties of the technique in those samples). Since in other species the centromere is the place where this protein is mostly located, if it is possible, the authors should perform a double immunolabelling with any ACA serum, to check it. Indeed, in platypus metaphase, the sister kinetochores seem to be the locations of some of the dots (it´s not so clear in echidna metaphase spermatocyte).
Specific comments
In Fig. 1: It would be great if they could add some small cohesin subunits along the metaphase chromosome arms, since they are not completely released until the end of metaphase. I see that there is not many space, but perhaps some really small dots.
In Fig 3: I cannot differentiate very well blue from green color.
In Pag. 9 239-241. They should clarify the information of this section, maybe they can specify which proteins are more expressed in meiosis
In line 316, it is explained that the increase of expression of CPC in ovaries and testes can be due to their additional meiotic functions on top of mitosis. I think that it is more probably due to the increased rate of division in those tissues, as they explain later for the increase of different proteins in ovaries.
In the Immunofluorescence images, in both figures 4 and 6, it would be appreciated if the name of the species appeared in the corresponding rows or columns.

Author Response
Dear Reviewer 1
We are grateful for the positive and constructive comments and we are pleased to submit a revised manuscript where we have taken on almost all the suggested changes (detailed below):
REVIEWER 1
As a general comment, once they demonstrated the absence of AURKC in both species, it would be desirable that they refer to it as AURKBC or explain why some times they refer to it as AURKB. I understand it for the immunolabelling, because they used an anti-AURKB antibody, but sometimes it is not very clear for me the difference. For example, in page 7 firs paragraph “Recent publication of the first echidna genome [18] enabled comparative analysis of monotreme AURKBC – this gene was mapped to echidna chromosome Y4. However, because AURKBC is likely critical for chromosome segregation in females too, we mapped 208 the echidna ovary transcriptome to the echidna genome and found that AURKB transcripts were also present in females. Moreover, platypus AURKB is located upstream of genes that have been previously assigned to the pseudo-autosomal region (PAR) of chromosome X5 (for example FLOT1), suggesting that platypus AURKB also localizes to this PAR. Because platypus PAR of X5 is syntenic to echidna PAR of X4 [18], this suggests that echidna AURKB would be located in PAR of Y4/X4. “
- We corrected the nomenclature – in all cases where we refer to monotremes, AURKB has been changed to AURKBC.
I understand how difficult it is to immunolabelling proteins in testes in those species and the great value of their results but, in my opinion the SMC3 results in echidna doesn´t have much to do with the other results. In fact, they haven´t discuss very much about it. I understand they use SMC3 to identify the different meiotic stages and that what they have found is an interesting result, but maybe it could be remove from the summary.
- We tried to make it clearer that the immunofluorescence comparison between platypus and echidna SMC3 is one of the major points of the manuscript and we think it would be a pity to leave this result out as it points to an important difference between relatively closely related species.
Related also with the immunodetection experiments, in relation with immunolabelling of AURKB in platypus and echidna, (Fig 4 C): In metaphase it is difficult to see which is the distribution of AURKB protein. The labelling appears as dots both on chromatin and on cytoplasm (maybe due to the difficulties of the technique in those samples). Since in other species the centromere is the place where this protein is mostly located, if it is possible, the authors should perform a double immunolabelling with any ACA serum, to check it. Indeed, in platypus metaphase, the sister kinetochores seem to be the locations of some of the dots (it´s not so clear in echidna metaphase spermatocyte).
- We agree with this accurate comment but sadly the majority of available antibodies do not cross-react with monotreme proteins and unfortunately the set of available centromeric antibodies is generally extremely limited, which prevents us from conducting a more in-depth analysis in this regard. We are happy to provide negative controls if needed.
In Fig. 1: It would be great if they could add some small cohesin subunits along the metaphase chromosome arms, since they are not completely released until the end of metaphase. I see that there is not many space, but perhaps some really small dots.
- The requested cohesin subunits along the chromosome arms have been added.
In Fig 3: I cannot differentiate very well blue from green color.
- We changed the blue and green color and made it darker so hopefully it is now easier to distinguish between them.
In Pag. 9 239-241. They should clarify the information of this section, maybe they can specify which proteins are more expressed in meiosis.
- We clarified which proteins are specific to meiosis, mitosis or both in the mentioned section and also in other sections throughout the manuscript.
In line 316, it is explained that the increase of expression of CPC in ovaries and testes can be due to their additional meiotic functions on top of mitosis. I think that it is more probably due to the increased rate of division in those tissues, as they explain later for the increase of different proteins in ovaries.
- We agree with this comment and changed the respective sentences so they reflect the increased rate of division in testes and ovaries rather than additional meiotic functions.
In the Immunofluorescence images, in both figures 4 and 6, it would be appreciated if the name of the species appeared in the corresponding rows or columns.
- We added species names to the corresponding rows or columns.
Reviewer 2 Report
The manuscript by Pajpach et al. analyzes the conservation, expression profiles and subcellular localization of a selection of genes involved in chromosome segregation in monotremes. An understanding of the evolution of factors involved in chromosome segregation is of great interest, and data from non-standard model organisms make a valuable contribution to it. The authors present alignments and synteny maps of AURK genes/proteins to confirm the lack of AURKC in monotremes. This evolutionary analysis of AURKs is interesting and well presented, although the figures miss some information, and the findings mainly appear to support previous findings. The authors then present expression profiles across different tissues for a selection of factors involved in chromosome segregation (focusing on the CPC, cohesin and KMN complexes). This analysis is based on previously published sequence datasets. For a few of these factors the authors also present IF images of either mitotic or meiotic cells.
Despite the interesting subject, this manuscript has severe limitations. The scope of the expression analysis presented is narrow, and it is unclear why it does not include a more complete set of genes involved in chromosome segregation (CENP-A, CCAN, kinetochore components are notably absent). Moreover, the description and interpretation of the expression profiles that are presented is superficial and in some instances not supported by the data. The significance of the findings therefore remains obscure. For example, it is very surprising that proteins that are supposedly in the same complex or at least act in the same process have markedly different expression patterns. Yet the results section skips over these discrepancies and does not give an explanation or interpretation. In the few instances where these results are interpreted, this is done in a way that leaves the reader with more questions than answers. The manuscript also falls short in the comparison of mitotic and meiotic functions of the factors analyzed. The immunofluorescence analysis is of interest, but does little to support the findings of the expression analysis, as the number of cell types and cell cycle stages shown is very limited. Moreover, controls for the antibody stainings are missing.
To improve the manuscript, I find it necessary that the authors present a more complete set of factors involved in chromosome segregation, including factors that only function in mitosis or meiosis as internal controls, and provide an in-depth comparison and discussion of these expression profiles. It would also be important to have a more complete IF analysis with the distribution of several members of the CPC, cohesion and KMN complexes in different cell types. This would be helpful to understand if the differences in expression profiles observed are biologically meaningful.
Additional points:
In the introduction, the authors raise a number of questions that are not addressed in the results section, while the actual aims of the study are only loosely described. E.g.
“However, it is currently unknown how this sex chromosome multiple faithfully segregates at meiosis and how this is controlled.” This is not at all addressed.
“However, very little is known about functions of AURKs when it comes to meiosis.” The authors detect AURK expression in mitotic tissues, but do not assess any functions.
It is ok to specify the open questions in the field, but it would be helpful if the authors could also more clearly formulate the questions that they are addressing in the results section.
Lines 98-99, the sentence should probably be: Our analysis of these key components in the basal lineage of monotremes confirmed that AURKA and AURKBC are present, but AURKC is not, providing further clarification…
Line 152, “Utilizing new genomic information…”. The authors should explain what is meant by new genomic information. Without this, it is difficult for the reader to understand how the analysis presented here constitutes an advance in our understanding of AURK evolution.
Lines 215-220. It is unclear where the transcription data is coming from and how it was analyzed. The source is referenced under data availability, but it is still not straightforward to understand the nature of the samples that were sequenced and how the data was derived. It would be important to have more insight into this, either here or in the methods section.
Lines 217-220. “Therefore, we quantified the expression of platypus AURKA and AURKB, showing that AURKBC transcript is preferentially produced in ovaries, testes and fibroblasts and AURKA transcript is produced on higher levels in all examined tissues with an unexpectedly high ovary expression (Fig. 4b)”
This sentence is misleading. Both proteins (not just AURKBC) are “preferentially produced in ovaries, testes and fibroblasts'', and while it is true that AURKA transcript levels are higher, they are not “produced on higher levels in all examined tissues”, but notably low in all tissues except ovaries, testes and fibroblasts. The authors should rephrase this part to reflect the data.
The argument comes back in the discussion (lines 284-286), where it is again misleading: “Indeed, the expression profile of platypus AURKBC suggests that this isoform is highly expressed in testes and ovaries (as opposed to more ubiquitous AURKB expression and gonad specific AURKC expression in eutherian mammals).”
It is correct that the data shows high AURKBC expression in testes and ovaries. However, it is also highly expressed in fibroblasts, and there is no data that would suggest that the observed AURKBC expression contrasts the more ubiquitous AURKB expression in eutherian mammals. In fact, the IF analysis the authors present is for mitotic, not meiotic cells.
The discussion part of the expression analysis (lines 314-329) contains several statements that are misleading or hard to interpret:
“CPC proteins (AURKB, BIRC5, CDCA8 and INCENP) were expressed throughout somatic tissues and had an increased expression in ovaries and testes, presumably because of their additional meiotic functions on top of mitosis”
None of these genes is expressed throughout somatic tissues, and INCENP does not show increased expression in ovaries and testes. The expression patterns are also remarkably different for genes encoding proteins that localize in the same complex.
“KNL1 and NDC80 expression patterns were similar to human or mouse.”
This may be true, although the comparison to human or mouse is not shown. More importantly, the two genes show very different expression patterns despite the fact that they act in the same complex, but this is not discussed.
“Cultured fibroblasts present a rapidly dividing cell population with a rapid turnover of chromosome segregation related proteins and therefore most proteins exhibited a dramatically high expression in this tissue, with the exception of KNL1, REC8, SMC1B and STAG3”
This is reassuring for the meiosis-specific factors, but why is KNL1 expression low in fibroblasts?
“In ovaries the expression was much higher than expected in case of AURKA, BIRC5, CDCA8, NDC80, SMC1B and SMC3, which may be due to the high division rate of the support cells in ovaries required for a successful oogenesis.”
Why is this unexpected if there is active cell division in ovaries, and why are INCENP and KNL1 not also highly expressed in ovaries in this case?
Part of the introduction, lines 55-63, is repeated almost verbatim in the discussion, lines 271-281.
The authors use sometimes AURKB and sometimes AURKBC to describe what appears to be the same monotreme gene. E.g. lines 217-218 “we quantified the expression of platypus AURKA and AURKB, showing that AURKBC transcript is preferentially produced”, or Figure 4. This is confusing.
In all figures with bar plots, the authors should also show the individual data points in addition to the mean and the s.d.
Figure 1, the metaphase drawing suggests that KMN is only present on one side of the spindle. This should be corrected.
Figure 2. For the middle panel, the genes that are not in the same position for each lineage are not always connected, making the map confusing. What is the gene underneath the label BORCS6 in monotremes? It is apparently not BORCS6.
Figure 3. The individual sequences are not annotated in the figure. The colors shown are different from the ones mentioned in the legend.
Figure 4C. The legend should mention the cell type show. It would be informative to describe the actual localization in addition to stating that it is similar to other mammals.
Figure 4C and 6. How are the authors sure that these antibodies are specific to the proteins they want to detect in monotremes? Do they have negative controls? Or can they at least show that the epitopes targeted by the antibodies are conserved?
Figure 4C and 6, what is the blue staining? I am assuming that this is DAPI counterstaining, but it is explained neither in the figure legends nor in the methods section.
Figure 5 should also include SMC1A and SA-1/2 as controls for mitosis-specific factors. It would also help if the factors found in the same complexes are clearly grouped.
Figure 6. The legend should describe the cell types shown in more detail.
It is surprising that the authors chose completely different presentation styles for assessing the conservation of AURK (Figure 3), SMC3 (Figure S1, where they only align monotreme sequences) and INCENP (Figure S2).
Author Response
Dear Reviewer 2,
thanks so much for the constructive and positive comments. We hope to have addressed the comments satisfactorily in our revisions.
REVIEWER 2
Despite the interesting subject, this manuscript has severe limitations. The scope of the expression analysis presented is narrow, and it is unclear why it does not include a more complete set of genes involved in chromosome segregation (CENP-A, CCAN, kinetochore components are notably absent). Moreover, the description and interpretation of the expression profiles that are presented is superficial and in some instances not supported by the data. The significance of the findings therefore remains obscure. For example, it is very surprising that proteins that are supposedly in the same complex or at least act in the same process have markedly different expression patterns. Yet the results section skips over these discrepancies and does not give an explanation or interpretation. In the few instances where these results are interpreted, this is done in a way that leaves the reader with more questions than answers. The manuscript also falls short in the comparison of mitotic and meiotic functions of the factors analyzed. The immunofluorescence analysis is of interest, but does little to support the findings of the expression analysis, as the number of cell types and cell cycle stages shown is very limited. Moreover, controls for the antibody stainings are missing.
To improve the manuscript, I find it necessary that the authors present a more complete set of factors involved in chromosome segregation, including factors that only function in mitosis or meiosis as internal controls, and provide an in-depth comparison and discussion of these expression profiles. It would also be important to have a more complete IF analysis with the distribution of several members of the CPC, cohesion and KMN complexes in different cell types. This would be helpful to understand if the differences in expression profiles observed are biologically meaningful.
- The provided suggestions and comments provide valuable advice on how to improve the manuscript. Therefore, we expanded the expression analysis by including the following genes: MIS12 as the final member of KMN complex (together with KNL1 and NDC80); mitotic cohesins SMC1A, STAG1 and STAG2; nucleosome and inner kinetochore proteins CENP-A, B and C; central kinetochore component CENP-S/T/W/X (excepting CENP-X as it is assigned to an unplaced scaffold); all members of the CCAN excepting CENP-R as it is not available for platypus. We revised the result and discussion sections related to the expression analysis, so it now provides better analysis and conclusions and so discussion reflects the observed data better, including unexpected differences and unusual expression patterns. We agree that expanding the immunofluorescence analysis would be of great benefit, but the presented antibodies are the only ones that cross-react with monotreme proteins – indeed, availability of antibodies that cross-react with the monotreme proteins under investigation is limited and a range of other antibodies have been tested by us and collaborators and do not cross-react with the monotreme proteins. We are happy to provide negative controls if needed.
In the introduction, the authors raise a number of questions that are not addressed in the results section, while the actual aims of the study are only loosely described. E.g.
“However, it is currently unknown how this sex chromosome multiple faithfully segregates at meiosis and how this is controlled.” This is not at all addressed.
“However, very little is known about functions of AURKs when it comes to meiosis.” The authors detect AURK expression in mitotic tissues, but do not assess any functions.
It is ok to specify the open questions in the field, but it would be helpful if the authors could also more clearly formulate the questions that they are addressing in the results section.
- We revised the mentioned sections, so it is now clearer what our manuscript tries to address and clearly distinguish this from the open questions in the field that we found to be important to mention as well.
Lines 98-99, the sentence should probably be: Our analysis of these key components in the basal lineage of monotremes confirmed that AURKA and AURKBC are present, but AURKC is not, providing further clarification…
- We agree and corrected the sentence accordingly.
Line 152, “Utilizing new genomic information…”. The authors should explain what is meant by new genomic information. Without this, it is difficult for the reader to understand how the analysis presented here constitutes an advance in our understanding of AURK evolution.
- We clarified this by specifying that this includes the newly assembled platypus and echidna genomes that became available in early 2021 (including the reference) – until then, only a limited platypus and no echidna genomes were available.
Lines 215-220. It is unclear where the transcription data is coming from and how it was analyzed. The source is referenced under data availability, but it is still not straightforward to understand the nature of the samples that were sequenced and how the data was derived. It would be important to have more insight into this, either here or in the methods section.
- We clarified this by expanding the description of how everything was derived and analyzed in the methods sections and linked it to the data availability section.
Lines 217-220. “Therefore, we quantified the expression of platypus AURKA and AURKB, showing that AURKBC transcript is preferentially produced in ovaries, testes and fibroblasts and AURKA transcript is produced on higher levels in all examined tissues with an unexpectedly high ovary expression (Fig. 4b)”
This sentence is misleading. Both proteins (not just AURKBC) are “preferentially produced in ovaries, testes and fibroblasts'', and while it is true that AURKA transcript levels are higher, they are not “produced on higher levels in all examined tissues”, but notably low in all tissues except ovaries, testes and fibroblasts. The authors should rephrase this part to reflect the data.
The argument comes back in the discussion (lines 284-286), where it is again misleading: “Indeed, the expression profile of platypus AURKBC suggests that this isoform is highly expressed in testes and ovaries (as opposed to more ubiquitous AURKB expression and gonad specific AURKC expression in eutherian mammals).”
It is correct that the data shows high AURKBC expression in testes and ovaries. However, it is also highly expressed in fibroblasts, and there is no data that would suggest that the observed AURKBC expression contrasts the more ubiquitous AURKB expression in eutherian mammals. In fact, the IF analysis the authors present is for mitotic, not meiotic cells.
- We re-worded and revised the mentioned sections so they better reflect the observed data and compare it better to what is available for other mammals.
The discussion part of the expression analysis (lines 314-329) contains several statements that are misleading or hard to interpret:
“CPC proteins (AURKB, BIRC5, CDCA8 and INCENP) were expressed throughout somatic tissues and had an increased expression in ovaries and testes, presumably because of their additional meiotic functions on top of mitosis”
None of these genes is expressed throughout somatic tissues, and INCENP does not show increased expression in ovaries and testes. The expression patterns are also remarkably different for genes encoding proteins that localize in the same complex.
“KNL1 and NDC80 expression patterns were similar to human or mouse.”
This may be true, although the comparison to human or mouse is not shown. More importantly, the two genes show very different expression patterns despite the fact that they act in the same complex, but this is not discussed.
“Cultured fibroblasts present a rapidly dividing cell population with a rapid turnover of chromosome segregation related proteins and therefore most proteins exhibited a dramatically high expression in this tissue, with the exception of KNL1, REC8, SMC1B and STAG3”
This is reassuring for the meiosis-specific factors, but why is KNL1 expression low in fibroblasts?
“In ovaries the expression was much higher than expected in case of AURKA, BIRC5, CDCA8, NDC80, SMC1B and SMC3, which may be due to the high division rate of the support cells in ovaries required for a successful oogenesis.”
Why is this unexpected if there is active cell division in ovaries, and why are INCENP and KNL1 not also highly expressed in ovaries in this case?
- We tried to address all of these comments by giving the result and discussion sections related to the expression analysis a considerable overhaul – we discuss the mentioned unexpected differences and unusual expression patterns in more detail and provide reasons that could explain it. We clarify that every comparison with human or mouse is done by utilizing publicly available transcriptome analysis on NCBI. In case of certain proteins, the y-axis scale representing RPKM is very different and makes it look like expression patterns of some proteins are different when they are actually similar, because a rapid increase in one tissue masks signals from other tissues. To address this, for the proteins where this is the case, we included inset charts that lack the “over-powering” tissue, so it is easier to look at finer expression in the rest of tissues. This was done by manually scaling y-axis.
Part of the introduction, lines 55-63, is repeated almost verbatim in the discussion, lines 271-281.
- We re-worder the mentioned sections so this is no longer tha case.
The authors use sometimes AURKB and sometimes AURKBC to describe what appears to be the same monotreme gene. E.g. lines 217-218 “we quantified the expression of platypus AURKA and AURKB, showing that AURKBC transcript is preferentially produced”, or Figure 4. This is confusing.
- We corrected this by referring to monotreme AURKB as AURKBC only everywhere in the manuscript.
In all figures with bar plots, the authors should also show the individual data points in addition to the mean and the s.d.
- We included the individual data points in all bar plots.
Figure 1, the metaphase drawing suggests that KMN is only present on one side of the spindle. This should be corrected.
- We added KMN dots to the other side of the spindle too.
Figure 2. For the middle panel, the genes that are not in the same position for each lineage are not always connected, making the map confusing. What is the gene underneath the label BORCS6 in monotremes? It is apparently not BORCS6.
- We clarified this by connecting the appropriate respective genes.
Figure 3. The individual sequences are not annotated in the figure. The colors shown are different from the ones mentioned in the legend.
- We added the annotations. We improved the colors, so they hopefully match the legend better now.
Figure 4C. The legend should mention the cell type show. It would be informative to describe the actual localization in addition to stating that it is similar to other mammals.
- The legend now contains the cell type shown, as well as better description of localization.
Figure 4C and 6. How are the authors sure that these antibodies are specific to the proteins they want to detect in monotremes? Do they have negative controls? Or can they at least show that the epitopes targeted by the antibodies are conserved?
- We are happy to provide the negative controls if needed – they do indicate that our antibodies are specific to the monotreme proteins. For SMC3, this antibody had been used in platypus previously in other studies and is generally very reliable with highly conserved epitopes (the sequence is provided by Abcam and shows 99% similarity with the respective monotreme SMC3 section which we can provide if needed). For AURKB and INCENP antibodies, these are much less conserved but reported to work in a number of different vertebrate species and again, we are happy to provide negative controls if needed.
Figure 4C and 6, what is the blue staining? I am assuming that this is DAPI counterstaining, but it is explained neither in the figure legends nor in the methods section.
- We clarified that blue signal is DAPI in both methods and figure legends.
Figure 5 should also include SMC1A and SA-1/2 as controls for mitosis-specific factors. It would also help if the factors found in the same complexes are clearly grouped.
- We added SMC1A, STAG1 (SA-1) and 2 grouped together proteins belonging to the same complexes.
Figure 6. The legend should describe the cell types shown in more detail.
- We added a more detailed description of the cell types shown.
It is surprising that the authors chose completely different presentation styles for assessing the conservation of AURK (Figure 3), SMC3 (Figure S1, where they only align monotreme sequences) and INCENP (Figure S2).
- We decided to remove the SMC3 alignment (as it was only showing that platypus and echidna SMC3 are identical, which is now only stated in a sentence) and replace it with the expanded expression analysis. For the remaining alignments (AURKs and INCENPs), they are different because they show different things – the comparison of domains in case of AURKs and evolutionary conservation in case of INCENPs.
Round 2
Reviewer 2 Report
The authors have addressed the majority of my concerns, have added additional data, and improved the clarity of their manuscript. I have no further comments.
Author Response
excellent we are very pleased that we have been able to address the comments and would like to thank you for your constructive comments.